# Synthesis and Characterization of Nano Fe and Mn (hydr)oxides to Be Used as Natural Sorbents and Micronutrient Fertilizers

María Teresa Cieschi [1], Marina de Francisco [1], Paula Herrero [1], Jorge Sánchez-Marcos [2], Jaime Cuevas [3], Elvira Esteban [1], Juan José Lucena [1] and Felipe Yunta [1,*]

1 Department of Agricultural Chemistry and Food Sciences, Autonomous University of Madrid, c/Francisco Tomás y Valiente, 7, 28049 Madrid, Spain; maria.cieschi@uam.es (M.T.C.); marina.dfh@gmail.com (M.d.F.); paula.herrero95@hotmail.commailto (P.H.); elvira.esteban@uam.es (E.E.); juanjose.lucena@uam.es (J.J.L.)

2 Applied Physics Department, Autonomous University of Madrid, c/Francisco Tomás y Valiente, 7, 28049 Madrid, Spain; jorge.sanchezm@uam.es

3 Department of Geology and Geochemistry, Autonomous University of Madrid; c/Francisco Tomás y Valiente, 7, 28049 Madrid, Spain; jaime.cuevas@uam.es

* Correspondence: felipe.yunta@uam.es; Tel.: +34-9014976265

**Abstract:** Fe and Mn (hydr)oxides are widely used as contaminant sorbents in water/wastewater systems but their potential use as micronutrient fertilizers is still poorly known. In this research, four nano-metal (hydr)oxides (amorphous Mn oxide (AMO), Fe-Mn binary oxide (FMBO), two-line ferrihydrite (2L-Fh) and goethite) were successfully synthesized and completely characterized (infrared and Mössbauer spectroscopy, X-ray diffraction particle size, specific surface area, point of zero charge). AMO, FMBO and 2L-Fh were introduced to interact with $AgNO_3$ (20.0 µM) and $TlNO_3$ (100.0 µM) diluted solutions for three days to check their potential capability as potential $Ag^+$ and $Tl^+$ adsorbents. AMO and FMBO (4% *w/w*) were tested as nanofertilizers by arranging a hydroponic bioassay for 35 days on white lupin culture as a Mn-hyperaccumulator plant model. AMO structure was identified as an amorphous mixture of Mn oxides while FMBO was an Fe dopped birnessite. Both materials were efficient in extracting $Ag^+$ and $Tl^+$ although large Mn concentration was released from FMBO to the solutions. AMO and FMBO promoted Fe and Mn nutrition in plants. Synthetic iron chelate (Fe-EDDHA), present in the nutrient dissolution, could be adsorbed onto AMO surface by producing Fe and Mn accumulation in roots and increasing Mn uptake rate without toxicity symptoms. Therefore, AMO and FMBO not only demonstrated their efficiency as adsorbents, but also displayed they would be promising nanomaterials as micronutrient fertilizers.

**Keywords:** nano-metal oxides; amorphous Mn oxide; Fe-Mn binary oxide; nanoadsorbents; nanofertilizers

## 1. Introduction

Iron and manganese oxides are the most studied nanomaterials (NMOs) and are of particular interest because they have strong redox and sorption properties that can limit the mobility of toxic metals within soils, even under a range of redox conditions [1]. However, there are few studies focused on these NMOs in cropping systems [2,3].

Among the most studied nano-iron oxides for heavy metals removal from water/waste water include amorphous hydrous Fe oxides (ferrihydrite) and goethite ($\alpha$-FeOOH). Ferrihydrite is much more reactive compared to other iron oxides, which is attributed to its high degree of structural disorder and high specific surface area [4]. It is a particularly effective sorbent for heavy metals as Cu, Cr and Pb [5] and arsenate [6] and it is used in wastewater treatment [7]. On the other hand, goethite, a crystalline stabilized oxy-hydroxide, shows strong affinities for surface binding of oxyanions and heavy metals [8] as Cd, Cu, Co, Pb and Zn [9] or phosphates [10] from wastewaters and industrial effluents. However, Mn oxides are often more effective than Fe oxides for the stabilization of some metal(loid)s, since the

formation of inner-sphere complexes results in strong bonds between the metal(loid)s and Mn oxide surface [3]. Poorly crystalline Mn oxides or amorphous Mn oxides (AMOs) are highly effective stabilizing amendments of metal(loid)s in water and soil [3,11,12]. AMOs have presented high sorption efficiencies for various metal(loid)s, such as Cu, Cd, Zn, Pb, and As from water and soil over a wide range of pH values. Nevertheless, one of the drawbacks in using AMO is Mn leaching at lower pH values and/or under reducing conditions. In addition, ferro-manganese binary oxide (FMBO) is a bimetallic oxide that contains iron and manganese. It has been used for metal ions removal from aqueous medium [13], in particular, As (III), As(V), Co(II), SB(V) and Tl(III). It has superior performance mainly because they combine in just one material the oxidation performance of manganese oxide and the high adsorption of iron oxide [14].

Nanoparticles (NPs) are defined as natural or engineered small particles with a size between 1 and 100 nm which, compared to their bulk counterparts, exhibit significantly different physical and chemical properties. [15]. Several studies [2,15,16] have evaluated the effect of engineered NPs in plant tissues to understand their interactions with the soil-plant system and to study unsuitable toxicity drawbacks. Effects of NPs on crop growth depends on the type, source, concentration in soil solution and size of the NPs, the plant species (development stage and growth rate), and the exposure time of NPs to crops. Besides NMO's ability in extracting metals from water/wastewaters, the NMOs size and surface charges make of them candidates for uptake by plants. Moreover, Fe and Mn are essential micronutrients that play important physiological processes in plants and their deficiencies are widely observed in crops grown in calcareous soils [17]. However, only a small handful of studies have investigated the contribution of NMOs in plant nutrition due to fear of potential toxic effects of NMOs on crops. Pariona et al. [2] observed that ferrihydrite (1, 2, 4 and 6 g L$^{-1}$) increased germination rate, growth and the chlorophyll content of maize seedlings. Moreover, clusters of ferrihydrite nano particles were found in endodermis, xylem, phloem vessels, and cell walls of the xylem vessels of maize stems. Michálková et al. [3] reported that 1% of synthetic AMO applied to contaminated soil reduced contaminants uptake and improved sunflower growth although plants presented Mn toxicity symptoms. Additionally, plant-NMOs interaction studies are lacking. Therefore, in this study, synthetics AMO and FMBO were evaluated as heavy metal absorbents and ecofriendly micronutrient fertilizers. To achieve this aim, four NMOs were synthetized (AMO, FMBO, ferrihydrite, goethite,), tested as emergent heavy metals (Tl$^+$ and Ag$^+$) adsorbents and their contribution in Fe and Mn nutrition for white lupin plants was evaluated.

## 2. Materials and Methods

### 2.1. Reagents

All reagents used were of recognized analytical grade, and solutions were prepared with type-I grade water [18] free of organic contaminants (Millipore, Milford, CT, USA).

### 2.2. Synthesis of Nano-Amorphous Metal Oxides

AMO was synthesized according to the reduction method, proposed by Luo et al. [19] and slightly modified. In brief, a solution containing $15.00 \pm 0.01$ g of $KMnO_4$ and $30.00 \pm 0.01$ g of KOH in 300 mL ultrapure water was added, with stirring to a mixture of 100 mL of ethanol (98% *v/v*) and 200 mL of ultrapure water with $30.00 \pm 0.01$ g of KOH. The resultant gel was left to age for 24 h at 20 °C and divided in two aliquots of 300 mL. One of these aliquots was stored to prepare the FMBO and the second one was repeatedly washed with ethanol (98% *v/v*) until the final conductivity was 35 μS·cm$^{-1}$. The aged suspension was freeze-dried and ground to obtain a black powder.

To synthesize the FMBO, 100 mL of 1.78 M $Fe(NO_3)_3$ was added to the remaining aliquot (300 mL) of the aged gel used to obtain the Mn oxide. The Fe:Mn molar ratio obtained was 3.76, implying that Fe and Mn oxides were homogeneously distributed, as already reported elsewhere [20–22]. The resulting sample suspension was allowed to age

for 24 h at 20 °C in the dark and then was repeatedly washed with ethanol (98% *v/v*) until the final conductivity was around 32 µS·cm$^{-1}$. The aged suspension was freeze-dried and ground to obtain a reddish-dark powder.

Two-line ferrihydrite (2L-Fh) was synthesized according to the procedure proposed by Schwertmann and Cornell [23]. Either 5 or 1 M NaOH solutions (depending on the curve zone) were slowly added to 0.4 M Fe(NO$_3$)$_3$·9H$_2$O up to a final pH of around 7.0 [24]. The resulting sample suspension was allowed to age for 24 h at 20 °C in the dark and then was repeatedly washed with ultrapure water until the final conductivity reached 22 µS·cm$^{-1}$. The aged suspension was freeze-dried and ground to obtain a reddish-brown powder. A similar procedure was followed for goethite synthesis [23]. In fact, NaOH volumes were slowly added to 0.4 M Fe(NO$_3$)$_3$ solution up to a final pH of 12. The resulting suspension was left to age for 72 h at 65 °C in the dark, and after several washes, the resulting dialyzed suspension was freeze-dried and ground to obtain a yellowish powder.

### 2.3. Characterization of Nano Metal Oxides

The nanomaterials synthesized were subjected to properly physicochemical characterization: functional groups identification by Fourier transform infrared spectroscopy (FTIR), crystallinity grade by X-ray diffraction (XRD), particle size estimation by Scherrer equation (1), specific surface area by BET (Brunauer, Emmett and Teller) method, point of zero charge (PZC) measurement, oxidation state of iron in 2L-Fh, goethite and FMBO by Mössbauer spectroscopy. The FT-IR spectra of a mixture of each NMO and KBr (1.0 mg of sample + 99.0 mg of dry KBr) from 4000 to 550 cm$^{-1}$ was recorded on a Bruker IFS66v FT-IR spectrophotometer fitted with an apparatus for diffuse reflectance. The XRD was performed on randomly oriented dried powder and oriented <2 µm size fraction. The diffractometer used was an X-PERT Analytical instrument with an X-CELERATOR detector. Measurements were taken at 0.016° 2θ angular steps for a time of exposure of 100 s per step. The equipment uses monochromatic radiation provided by a Ge 111 monochromator. The software X'Pert Highscore was used for data treatment and identification of mineral phases. The Scherrer Equation (1) applied to determine the crystallite size of polycrystalline samples is a widely used tool [25], where λ, *B*, θ, and *K* (0.91) are the X-ray wavelength (0.154 nm), full width at half-maximum (FWHM), corresponding Bragg angle and Scherrer constant, respectively.

$$D = \frac{K\,\lambda}{B\,\cos\theta} \tag{1}$$

The specific surface area of the solid NMOs was measured by using the BET-method by N$_2$ adsorption with a Gemini V analyzer from Micromeritics™. The point zero of charge (PZC) was determined by potentiometric mass titration. Three solutions from each NMO were prepared by dissolving 0.2000 ± 0.0001 g in 50 mL of ultrapure water at different NaNO$_3$ ionic strength levels (0.5 M, 0.05 M and 0.005 M). An aliquot (20 mL) of each solution was titrated with NaOH 1.0 M up to reach pH > 10 and other aliquot (20 mL) was titrated with HNO$_3$ 0.1 M up to reach pH < 4. The PZC was determined by difference between H$^+$ added and H$^+$ resting in solution. Then, the PZC represents H$^+$ absorbed by the NMOs.

Mössbauer spectra were recorded in triangular mode using a conventional spectrometer with a 50-µCi $^{57}$Co (Rh) gamma-ray source mounted on an electromagnetic transducer in transmission mode. The data obtained at room temperature was analyzed with the NORMOS program [26]. The spectrometer was calibrated for each velocity using an alpha-Fe (6 µm) foil.

### 2.4. Heavy Metal Interactions and Nutrients Releasing

In order to test the NMOs behavior as potential heavy metal sorbents, 0.1000 ± 0.0001 g of each 2 L-Fh, AMO or FMBO solid was introduced to interact with 10.0 mL of TlNO$_3$ (100.0 µM) or AgNO$_3$ (20.0 µM) in 15.0 mL polyethylene flasks by triplicates. The flasks were shaken at 120 m$^{-1}$ for 24h at room temperature. Suspensions were centrifuged at

4000 m$^{-1}$ for 5 min (Sorvall Legend XFR, Thermo Fisher Scientific, Waltham, MA, USA). Supernatants were filtered through a Millipore 0.45 µm membrane filter to determine both Ag and Tl dissolved concentrations by inductively coupled plasma mass spectrometry (ICP MS; NexION 300XX, PerkinElmer Inc, Madrid, Spain). The pH values were measured in the supernatants by using an Orion Research Ion Analyzer EA920 (Orion Research Inc., Boston, M.A., USA).

### 2.5. Plant Experiment

The white lupin (*Lupinus albus* cv. Marta) plant was chosen for this experiment as a Mn-hyperaccumulator plant model (up to 20 g Mn kg$^{-1}$ leaf dry weight) [27,28]. The seeds were germinated in the dark at 28 °C (Incubator Model IH-150) on perlite moistened with distilled water and sprayed with 10.0 mL of CaSO$_4$ 1.0 mM. After germination (4 days), uniform seedlings were placed on containers filled with 1/10 diluted nutrient solution. The composition of the nutrient solution (NS) was as follows: (macronutrients in mM) 3.0 Ca (NO$_3$)$_2$·H$_2$O, 3.0 KNO$_3$, 2.0 MgSO$_4$·7H$_2$O, 1.5 KH$_2$PO$_4$, (micronutrients in µM) 35.9 FeEDDHA, 1.6 ZnSO$_4$·7H$_2$O, 32.8 MnSO$_4$·H$_2$O, 1.6 CuSO$_4$·5H$_2$O, 1.0 (NH$_4$)$_6$·Mo$_7$O$_{24}$·4H$_2$O and 23.1 H$_3$BO$_3$. The nutrient solution pH was adjusted at 5.5–6 to reproduce agronomic conditions. After two days, the seedlings were transferred to 1.0 kg pots filled with 8.0 mm standard silica sand (CEN DIN EN 196-1) mixed with the following amendments: (1) 4.0% of FMBO, (2) 4.0% AMO. Control pots were prepared by using only silica sand. An on-demand plant nutrition system was carried out, allowing plants to take up the NS by capillarity. The NS containers were filled with NS complete which was replaced every week. The implemented design was four plants per pot and three replicates (three pots) per amendment. First sampling was performed at 35 days after NMOs application. Plants grew up during five weeks under controlled climatic conditions: day/night photoperiod, 13/11 h; temperature (day/night) 20/18 °C, relative humidity (day/night) 40/60%, into a Dycometal-type CCK growth chamber (Dycometal Equipos Control de Calidad, S.L. Barcelona, Spain).

### 2.6. Plant Material

The sampled shoots (leaves and stems) and roots were weighed, washed with ultrapure water, and dried in a forced air oven (Memmert Loading Model 100–800) at 70 °C for 3 days, weighed and finally milled by using a ball mill (Retsch Model MM 301) for 1 min at 16 s$^{-1}$. Plant grounded (0.3000 ± 0.0001 g) was digested by autoclave (Presoclave III P Selecta) with 4 mL of HNO$_3$ (65% *v/v*) and 1 mL of H$_2$O$_2$ (33% *v/v*) under a pressure of 1.5 kg cm$^{-2}$ for 30 min [29]. The concentrations of Fe and Mn in plant digests were analyzed by ICP-MS.

### 2.7. Statistics

A completely randomized design was developed with three factors (metal concentration × contaminant × NMOs) for the heavy metal batch experiment and two factors (micronutrient content × NMOs) for the plant experiment. Homogeneity of variance was firstly analyzed through Levene's test. One-way ANOVA and post hoc Tukey's HSD ($p < 0.05$) test were run to find out those differences between NMO groups. All the statistical calculations were performed using SPSS v.26.0$^{®}$ software.

## 3. Results

### 3.1. Characterization of Nano-Amorphous Metal Oxides

3.1.1. Infrared Spectroscopy

The FTIR is an efficient tool to verify the synthesis quality of NMOs, identifying the available functional groups on their structures at the end of synthesis. The FTIR spectra (4000 to 550 cm$^{-1}$) of NMOs are presented in Figure 1. Broad band between 3100–3400 was observed for all NMOs, corresponding to OH stretching vibrations arising from structural OH and adsorbed water [30]. Moreover, in the region between 1800 and 1300 cm$^{-1}$, diverse prominent

bands were observed: the peaks around 1800 $cm^{-1}$ were assigned to the presence of $CO_3^{2-}$ due to contamination by atmospheric $CO_2$, bands at 1620–1650 $cm^{-1}$ correspond to $H_2O$ deformation vibrations, bands at around 1450 $cm^{-1}$ were assigned to $\nu C-O$ stretching vibrations of carbonate adsorbed as hydrogen-bonded outer sphere species, while the sharp peak at 1384 $cm^{-1}$ was due to the $NO_3^-$ [5,10,31]. With respect to the 2L-Fh (Figure 1A), Fe–O lattice stretching mode appears at 589 $cm^{-1}$ [5,31]. In reference to goethite (Figure 1B), the Fe-O-OH bending bands at 892 ($\delta$-OH) and 795 $cm^{-1}$ ($\gamma$-OH) [5,10,32] which vibrate in and out, respectively, of the plane are important diagnostic bands and provide information about crystallinity. According to Cornell and Schwertmann [4], the bending bands also move closer together and a separation between them of 97 $cm^{-1}$ denotes a well crystallized goethite. Moreover, the symmetric Fe-O stretch at 636 $cm^{-1}$ is influenced by particle morphology and crystallinity. In this study, this band also confirm the synthesis of a well crystallized goethite. The band at 555 $cm^{-1}$ is related to an antisymmetric Fe-O vibration. Concerning AMO (Figure 1C), the O-H bending vibration combined with those of Mn atoms was assigned to the peak at 1561 $cm^{-1}$ [33]. The peaks from 1100 to 1000 $cm^{-1}$ indicated little hydroxyl group on AMO surfaces [34]. The broad band at 606 $cm^{-1}$ is attributed to the Mn-O lattice vibrations [12,33,35]. For the FMBO (Figure 1D), two peaks, at 1093 and 1050 $cm^{-1}$, respectively, were due to the bending vibration of the hydroxyl group associated with Fe and Mn on the surface of the FMBO [20,21,34]. The vibration peaks at 850 and 550 $cm^{-1}$ reflected the lattice vibrations (Mn-O and Fe-O) [21,36]. In particular, the broad band at 595 $cm^{-1}$ was due to the presence of Fe-O-Mn linkage in manganese-substituted iron hydroxide [37]. Furthermore, the FMBO spectrum denotes similarities with the 2L-Fh and the AMO spectra which would indicate the presence of both amorphous metal oxides in the structure of the FMBO [38].

### 3.1.2. Crystallinity

The XRD analysis of synthetic nano-oxides was undertaken to provide mineral phase confirmation and then, the XRD patterns are presented in Figure 2. In 2L-Fh pattern (Figure 2a) two typical peaks at 35 and 62 in $2\theta$ with poorly defined reflections were observed, coincident with 2-lines ferrihydrite reported by Chappell et al. [39] and Smith et al [7]. According to Figure 2b, goethite presented high crystallinity and the known principal peaks at 18, 21, 26, 34, 35 and 36 in $2\theta$, which were widely described by Schwertmann and Cornell [23]. For the AMO pattern (Figure 2c), two broad peaks were detected at 34 and 63 in $2\theta$ and assigned to an amorphous Mn oxide or hydrous Mn oxide, coincident with the description reported by Ouředníček et al. [33]. Moreover, three peaks well defined at 19, 24 and 70 in $2\theta$ were observed, which sugested that the AMO was undergoing a cristalinity process, maybe as a mixture of manganese oxides. In fact, the peak at 19 in $2\theta$ is characteristic for $Mn_2O_3$ and the peak at 24 in $2\theta$ is commonly observed for $Mn_3O_4$ [40]. Finally, for the FMBO pattern (Figure 2d), four broad peaks at 13, 25, 38 and 65 in $2\theta$ were detected, coincident with a birnessite structure reported by Di Leo et al. [35] and Gao et al. [41].

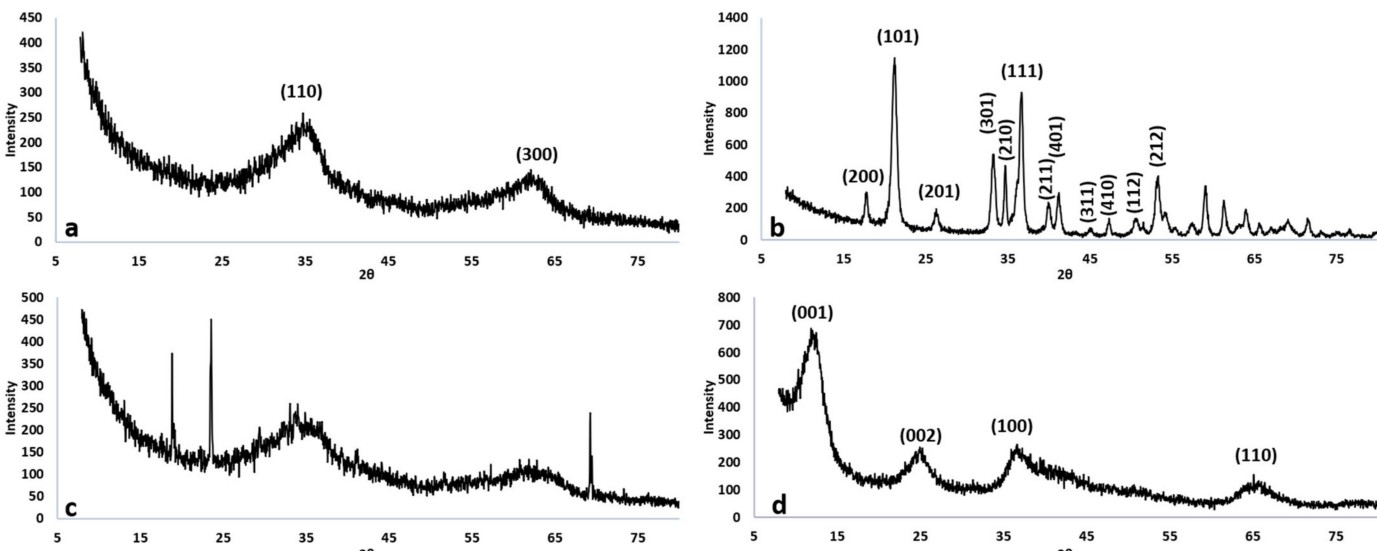

**Figure 1.** Infrared spectra of 2L-Fh (**A**), goethite (**B**), AMO (**C**) and FMBO (**D**).

**Figure 2.** X-RD patterns of (**a**) ferrihydrite, (**b**) goethite, (**c**) AMO and (**d**) FMBO.

### 3.1.3. Particle Size

In order to confirm that the metal oxides synthetized were NMOs, their particle sizes were calculated by using the Scherrer Equation (1), and the results are presented in Table 1. For 2L-Fh, the particle size obtained was less than the typical size of 2-lines ferrihydrite particles (2–5 nm) [39]. For goethite, the crystallite sizes resulted from the principal

crystallographic planes, which were in line with the results reported by Valezi et al. [42]. With respect to the AMO, the particle size was calculated considering the three peaks that denote crystalline forms, since these particles would have been the largest in the material structure. The result obtained is coincident with Kuo et al. [43] for commercial $Mn_2O_3$. Finally, the particle size estimated for FMBO corresponds to an amorphous material where de Fe:Mn ratio is higher to 3. According to Yin et al [44] and Lu et al. [45], the higher the Fe:Mn ratio in an iron doped birnessite, the smaller the particle size will be.

**Table 1.** Particle size, specific surface-area and point of zero charge [a].

| Nanomaterial | Crystallographic Plane | Size (nm) | BET ($m^2\ g^{-1}$) | PZC |
|---|---|---|---|---|
| 2L-Fh | (110) | 1.20 | $297 \pm 1.9$ | 8.00 |
|  | (300) | 1.44 |  |  |
| Goethite | (200) | 18.5 | $79.1 \pm 0.4$ | 8.10 |
|  | (101, 201, 301, 401) | 25.9 |  |  |
|  | (111, 211) | 18.9 |  |  |
| AMO | Peaks at $2\theta = 19°$, $24°,70°$ | 39.1 | $118 \pm 0.9$ | 9.80 |
| FMBO | (002, 100, 110) | 2.81 | $396 \pm 1.8$ | 6.20 |

[a] Results are expressed as averages $\pm$ standard error.

### 3.1.4. Specific Surface-Area

In Table 1 the BET surface-area values obtained for the synthetic NMOs are presented. The result obtained for 2L-Fh was coincident with Schwertmann and Cornell [23] which reported that BET surface-area for 2-lines ferrihydrite falls in the range of 200–300 $m^2\ g^{-1}$. The surface-area obtained for goethite was in line with the results reported by Tan et al. [46] and Jaiswal et al. [10]. The AMO's surface area observed was consistent with other results reported [43,47,48] for amorphous manganese oxides. Finally, FMBO solid showed the highest surface area, which it was higher than that reported by Zhang et al. [22].

### 3.1.5. Point of Zero Charge (PZC)

The PZC for the NMOs studied is presented in Figure 3. The PZC of synthetic Fe-oxides is well documented [6,19,20,23–25,39] and usually ranges between pH 7 and 9. Thus, PZC values for both, 2L-Fh and goethite was obtained to be 8 and 8.1, respectively. The PZC for AMO was measured to be near to 10, a higher value for those manganese oxides already reported [46,49,50] except for groutite ($\alpha$ MnOOH) [51]. Finally, the PZC obtained for the FMBO was 6.2, coincident with results obtained by Lu et al. [45] and An et al. [52].

### 3.1.6. Mössbauer Spectrometry

Figure 4 depicts the spectra for 2L-Fh, the FMBO and goethite at room temperature. 2L-Fh and FMBO samples (Figure 4a,b) show a spectrum with only a doublet. The fitting of both spectra has a very similar quadrupole splitting, 0.707 mm and 0.685 mm $s^{-1}$, respectively, and isomer-shift, 0.313 mm and 0.349 mm $s^{-1}$, respectively. These results agree with those previously reported for 2-lines ferrihydrite [2,53]. Yang et al. [38] also detected ferrihydrite in the FMBO structure when the molar ratio Fe:Mn was superior to 3. Conversely, the goethite sample, Figure 4c, shows a sextet expected for a magnetic ordered iron. However, the peaks are not symmetrical and a fitting with only one hyperfine magnetic field does not yield good results. This could be related to the at room temperature sample being near to the magnetic transition. For that reason, a distribution magnetic field has been used to fit the spectra. The data obtained in the fir, isomer-shift = 0.246 mm $s^{-1}$, quadrupole = −0.27 mm $s^{-1}$ and the average field of 37 Tesla, are very similar to the obtained by Forsyth et al. [54].

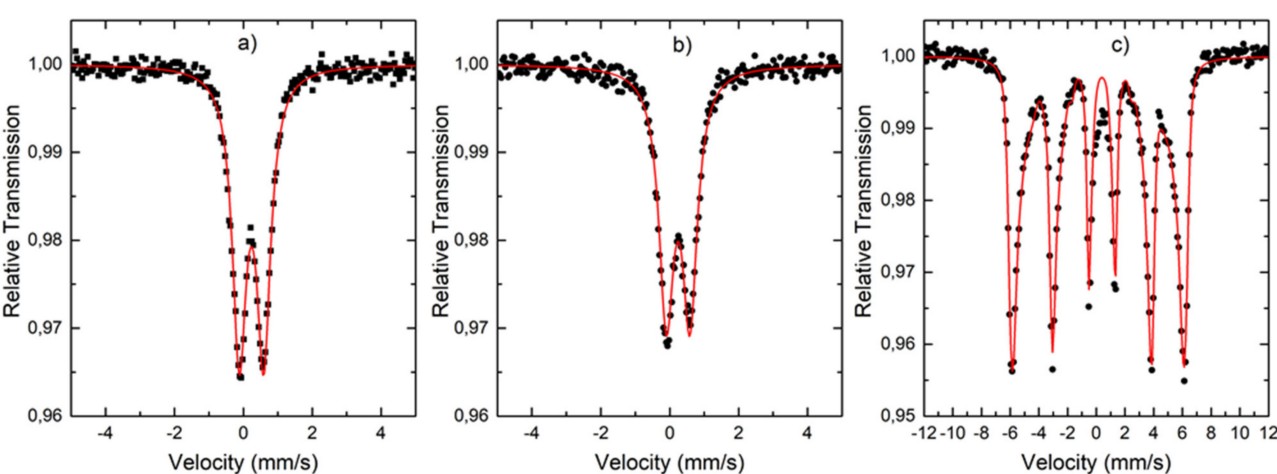

**Figure 3.** PZC of 2L-Fh (**A**), Goethite (**B**), AMO (**C**) and FMBO (**D**). Three ionic force was assayed: 0.005, 0.05 and 0.5 M.

**Figure 4.** Mössbauer spectra of ferrihydrite (**a**) FMBO (**b**) goethite, and (**c**) samples taken at room temperature. Solid red lines are the best fits and black dots indicate the experimental data.

### 3.2. Heavy Metals Interactions and Nutrients Releasing

With the aim of testing the NMOs' behavior as heavy metal sorbents, batch experiment was carried out by using both $NO_3Ag$ (20.0 μM) and $NO_3Tl$ (100.0 μM) diluted solutions

For this experiment, 2L-Fh, AMO and FMBO were chosen as they showed those largest specific surface-areas. Percentage of sorbed contaminants onto NMOs at several pH levels after three interaction days are presented in Figure 5. The pH of the $AgNO_3$ and $TlNO_3$ solutions before the interactions were 5.55 and 6.91, respectively. Final pH in the supernatants has changed, depending on the NMO to be 4.16 for 2L-Fh, 9.35 for AMO and 5.45 for FMBO. Only pH values for 2L-Fh suspension decreased significantly, while AMO and FMBO remained closer to their $PZC_{pH}$. No significant differences were observed for $Ag^+$ adsorption, indicating that all the tested NMOs were able to sorb this heavy metal at 100% of efficiency. Regarding to 2L-Fh results, it was coincident with Davis and Leckie [55], who reported that ferrihydrite exhibit a metal-characteristic sorption edge, beginning at about pH4 and reaching a maximum at pH8, reflecting its weak hydrolysis behavior and its general lower stability in bonds with oxygen electron donors. With respect to the results obtained for AMO and FMBO materials, their high adsorption efficiency may be explained by their high stability showed in the low variability of their $PZC_{pH}$ and their high specific surface-area.

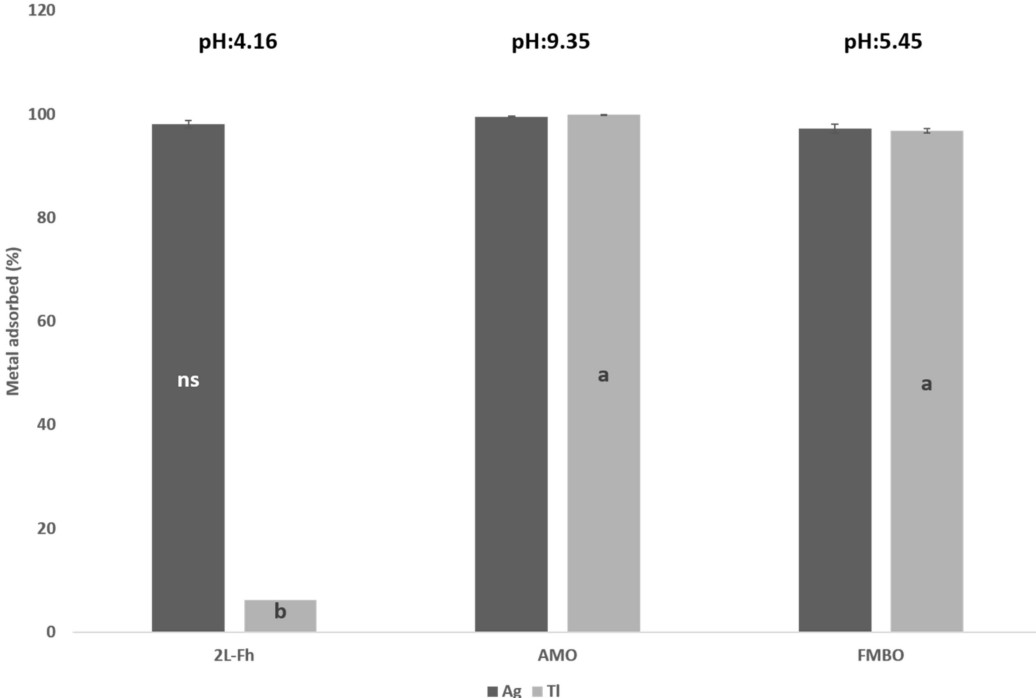

**Figure 5.** Content of $Ag^+$ and $Tl^+$ adsorbed (%) onto 2L-Fh, AMO and FMBO, and the final pH (upper part) after three days of interaction of $NO_3Ag$ (20.0 μM) and $NO_3Tl$ (100.0 μM) solutions with nano-amorphous metal oxides. For each series different letters denote significant differences among nanomaterials according to Tukey HSD Test ($p < 0.05$). ns: no significant differences.

Concerning to the $Tl^+$ adsorption, AMO and FMBO retained the highest content of $Tl^+$ in their structures. However, no 2L-Fh showed to be an efficient $Tl^+$ adsorbent material as the final interaction pH value (4) was out of the optimal adsorption pH range (5.5–11.0) as reported elsewhere [56]. However, the AMO and FMBO materials were efficient as $Tl^+$ adsorbents because their pHs were closer to their $PZC_{pHs}$ and within of that optimal sorption pH range. Indeed, Jacobson et al. [57] reported that high amounts of $Tl^+$ were sorbed onto birnessite (30% by mass), while it was relatively poorly sorbed on ferrihydrite at pH 5.1 (1.5% by mass), concluding that the pH was the main parameter responsible for their results. On the other hand, Li et al. [21] reported that the Fe-Mn binary oxide obtained by using a chemical oxidation and precipitation method, was an efficient $Tl^+$ adsorbent under a broad operating pH range (3–12).

Dissolved Fe and Mn concentrations after the interaction were plotted in Figure 6. No significant differences were observed among the contaminant solution used; then, each nanomaterial reacted in the same way for both heavy metals. However, 2L-Fh was the metal (hydr)oxide that released the highest content of Fe in solution, while FMBO had a similar behavior with respect to Mn content. According to these results, FMBO could work in both ways in adsorbing Tl in aquatic systems and in releasing Mn to be used for the crops as fertilizer under hydroponic conditions.

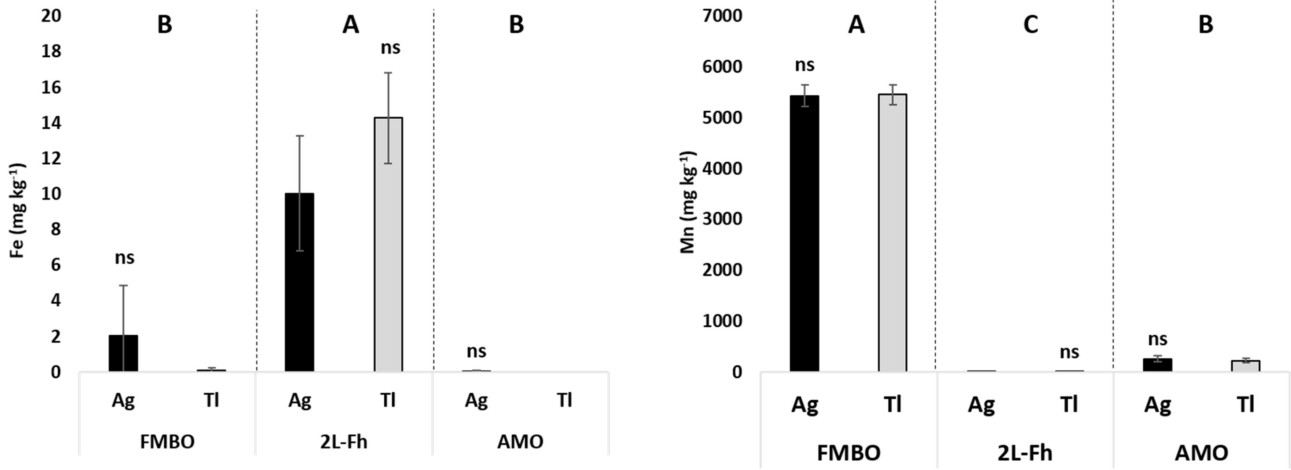

**Figure 6.** Dissolved Iron (**left**) and manganese (**right**) concentration (mg kg$^{-1}$) after three days of interaction of solutions of NO$_3$Ag and NO$_3$Tl with nano-amorphous metal oxides (FMBO, 2L-Fh, AMO). For each series different lowercase letters denote significant differences among the contaminants with respect to one nanomaterial, and uppercase denotes significant differences among nanomaterials according to Tukey HSD Test ($p < 0.05$). ns: no significant differences.

### 3.3. Plant Experiment

A hydroponic bioassay was conducted to evaluate the potential agricultural use of nano-amorphous metal oxides as micronutrient fertilizers. Both, AMO and FMBO materials were chosen as they are usually applied as heavy metal adsorbents but their contribution in plant nutrition was hardly ever studied. They were applied to pots where white lupin plants were grown. White lupin plants were chosen as a Mn hyperaccumulator plant model (up to 20 g Mn kg$^{-1}$ leaf dry weight) [28]. No significant differences in respect to the control plants for shoot dry weigh (control: 1.71 $\pm$ 0.21 g plant$^{-1}$, AMO: 1.63 $\pm$ 0.05 g plant$^{-1}$, FMBO: 1.88 $\pm$ 0.25 g plant$^{-1}$) or root dry weigh (control: 0.75 $\pm$ 0.06, AMO: 0.81 $\pm$ 0.14, FMBO: 1.13 $\pm$ 0.23) were found, showing that plants look well nourished. Both Fe and Mn contents in shoots and roots of white lupins plants after 35 days of NMOs application are shown in Figure 7. With respect to the Fe, it was majorly accumulated in white lupin roots when AMO was applied. A feasible explanation for these results can be found when Figure 8 is observed. Roots treated with AMO (Figure 8c) were reddish-orange color at the end of the bioassay. FeEDDHA surface sorption onto AMO could easily explain the reddish color of the roots.

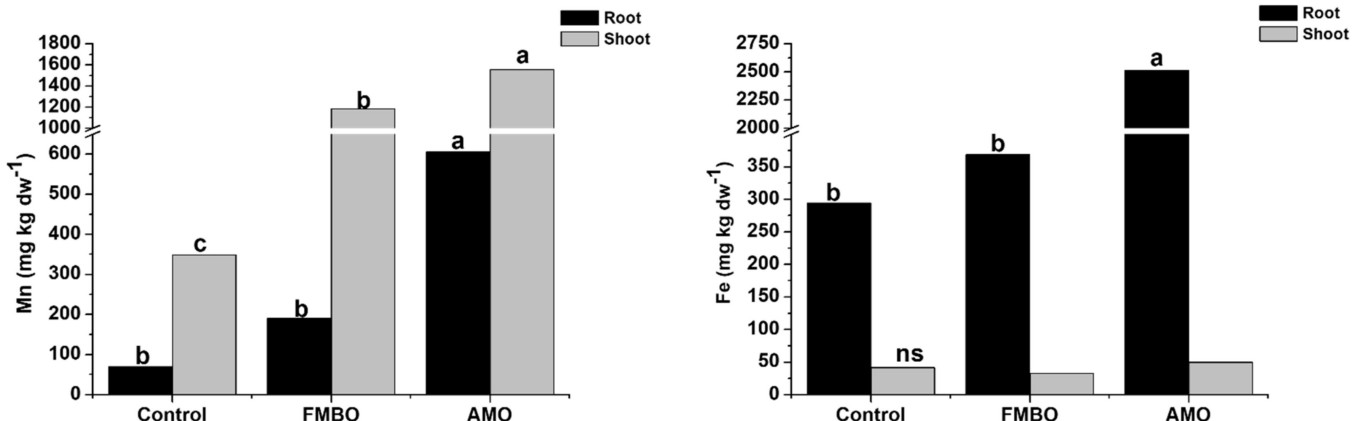

**Figure 7.** Iron (left) and Mn (right) contents (mg kg dw$^{-1}$) in shoot and roots of white lupin plants after 35 DAT. For each series different letters denote significant differences among the NMOs according to Duncan's Test ($p < 0.05$).

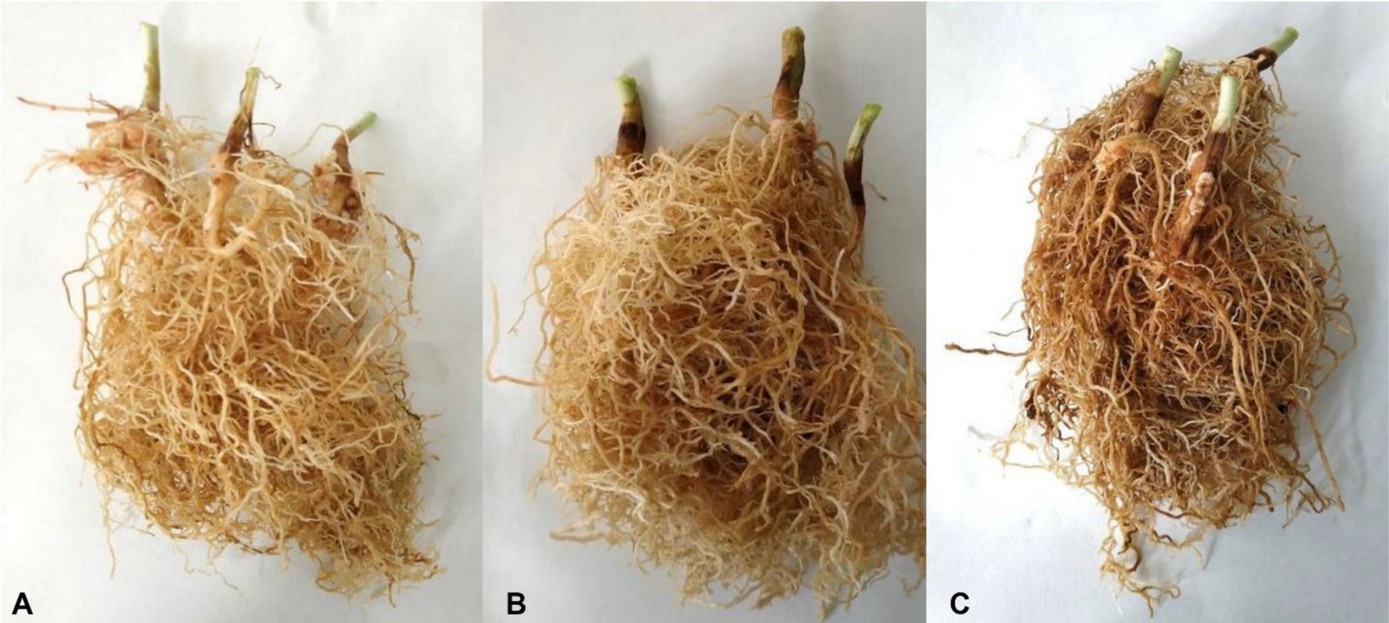

**Figure 8.** Roots of control white lupin plants (**A**) plants treated with FMBO (**B**), and plants treated with AMO (**C**) after 35 days of NMOs application.

Moreover, both AMO and FMBO nanomaterials showed a higher content of Mn in shoot than in control plants. Indeed, plants treated with AMO presented the highest content in root and shoot. It seems likely the EDDHA chelating agent released $Fe^{3+}$ over the AMO surface, chelated Mn from AMO and so, increasing Mn uptake. In any case, plants did not present Mn toxicity symptoms when they were grown at pH 5.5–6. Blamey et al. [28] studied the distribution and speciation of Mn in fresh roots, stems, and leaves of four crop species, including white lupin where the maximum dose applied was 100µM and white lupin plants did not present Mn toxicity symptoms. Michálková et al. [3] studied the influence of AMO application on the mobility of metals in bulk soil and sunflower (*Helianthus annuus* L.) rhizosphere together with their effects on plant growth and metal uptake. They reported that in acidic soils, AMO released excessive Mn concentrations and then, sunflower plants manifested symptoms of Mn phytotoxicity. Therefore, the authors recommended AMO as an amendment for soils with high pH and high cation exchange capacity. Concerning to FMBO, no Fe or Mn accumulation in roots was observed and despite the presence of ferrihydrite in its structure, no iron-manganese plaque developed over the roots. Then, significant differences in Mn content in shoots were observed respect

to the control plants and plants treated with AMO. The iron dopped birnessite structure of FMBO seems to be more stable than the amorphous structure of AMO, which implies a progressively Fe and Mn translocation from root to shoot. Hence, FMBO and AMO may would be an alternative for Mn nutrition in calcareous soils.

## 4. Discussion

Four NMOs (2L-Fh, goethite, AMO and FMBO) were synthetized and properly characterized. FTIR [4,31], XRD [39,42] and Mössbauer spectroscopy [2,53,54] confirmed that 2L-Fh and goethite were well synthesized, with adequate particle size [39,42], PZC [6,19,20,23–25,39] and specific surface area [9,46]. Conversely, the AMO XRD pattern (Figure 2c) confirmed that an amorphous Mn oxide was obtained, and it was undergoing a crystallinity process, as a mixture of manganese oxides [33,40]. Indeed, it presented a high specific surface, similar to that obtained by other authors [43,47,48] and a high PZC (pH:10), rarely found for Mn oxides [51]. The FMBO obtained was an iron doped birnessite [35,41] with a high specific surface area, consistent with an Fe:Mn ratio higher than 3 (3.755) [44,45] and it presented ferrihydrite in its structure (Figure 4b). Their surface morphology, crystal structure and $PZC_{pH}$ also play an essential role in pollutant removal, and these factors are directly affected by the Fe:Mn ratio in FMBO. The proper addition of Mn promotes the formation of amorphous iron oxide, which has greater affinity for contaminants due to it having more active sites and greater surface activity than the high crystalline goethite and lepidocrocite [37]. Finally, FMBO presented the highest specific surface area and so, showing a synergic effect between the Fe and Mn amorphous oxides, higher than that obtained for other authors [21], but this is attributed to differences in the synthesis method as reagents and aging time.

With the purpose of testing the NMOs behavior as potential heavy metal sorbents, 2L-Fh, AMO and FMBO were chosen since they have had the highest specific surface-area and, in consequence, they would be the most reactive and the most efficient nano-amorphous materials in scavenging pollutants. Silver and thallium were chosen because they are emergent contaminants due to the increase in their industrial release into the environment in recent years [57]. The strong oxidative activity of Ag-NPs releases silver ions, which results in several negative effects on biological systems by inducing cytotoxicity, genotoxicity, immunological responses, and even cell death [58]. According to Chen et al. [48], with the rapid development of economy, a large sum of Tl enters the aquatic environment from mining, metal smelting, industrial production, geothermal development, and electronic products etc., which aggravates the Tl pollution in the environment. Consequently, researching NMOs as heavy metals sorbents is highly necessary.

For this study, 2L-Fh was an efficient adsorbent for $Ag^+$, while it did not show the same behavior for $Tl^+$. The 2L-Fh exhibit a metal-characteristic sorption edge for $Ag^+$, beginning at about pH4.0 and reaching a maximum at pH8 [55], while the sorption of $Tl^+$ begins at about pH 5.5 and finishes around pH 11 [56]. The final pH for 2L-Fh at the interaction experiments was around 4, so this material was capable of retaining 100% of $Ag^+$ but only 6% of $Tl^+$. Moreover, 2L-Fh released high quantities of Fe to both polluted solutions due to the low stability associated with the weakly bonds with oxygen electron donors at pH4 [55]. With respect to AMO and FMBO materials, their high adsorption efficiency in retaining $Ag^+$ and $Tl^+$ is explained by their high stability, evidenced by the low variability of their $PZC_{pH}$ and their high specific surface-area. For this study, the retention of both pollutants onto AMO was almost 100%, superior to previous reports for $Tl^+$ adsorption [57] and similar for $Ag^+$ adsorption [59,60]. Concerning FMBO, it was an efficient adsorbent for both pollutants and its structure of iron dopped birnessite increased its adsorption capacity in comparison to their precursors, indeed, the 2L-Fh. In addition, FMBO released relevant contents of Mn to both polluted solutions because the equilibrium concentration would be superior to the pH obtained at the end of the interactions (pH: 5.45) [48]. Although in recent years FMBO has been tested as $Tl^+$ adsorbent [20,48], a scarcity of literature was found regarding

the specific use of this material or AMO in scavenging $Ag^+$ [58]. Additionally, AMO and FMBO would be adequate materials to adsorb $Ag^+$ and $Tl^+$ from waters and wastewaters.

Since AMO and FMBO are usually applied as heavy metal adsorbents but their contributions in plant nutrition were hardly ever studied [1], a hydroponic bioassay was conducted by applying those tested materials to pots where white lupin plants were grown. The Fe and Mn uptake was evaluated 35 days after nano-amorphous materials application. AMO produced iron and manganese accumulation (Figure 8) on the white lupin roots as the result of the interaction between AMO an FeEDDHA, which did not avoid Fe and Mn mobility to shoots (Figure 7) and did not produce toxicity symptoms. White lupin is a Mn accumulator plant, which tolerates up to 20 g Mn $kg^{-1}$ leaf dry weight and it can translocate $Mn^{2+}$ from the apoplast for storage in the vacuole, possibly via a Ca transporter, limiting the accumulation of Mn in the cytoplasm and cell wall [28]. The toxicity of Mn occurs in acid or waterlogged soils and AMO can release excessive Mn concentrations in acidic conditions [12], producing Mn toxicity symptoms in plants as sunflowers [3]. According to Rizwan et al. [16], NPs can pass through the plants by adhering to the root surfaces and entering the epidermis and cortex via an apoplastic pathway. However, uptake and translocation of NPs in plants may vary with plant species, cultivars, and growth conditions. Given this, it would be recommendable to apply AMO in calcareous conditions and test other plant species.

The FMBO obtained in this study is an iron dopped birnessite, where iron is part of its structure as ferrihydrite. The combination of Fe and Mn amorphous oxides changes the morphology and the characteristic structure of the final material [48]. Additionally, the FMBO seems to be more stable than AMO which would be under a crystallinity process, as a mixture of manganese oxides. Due to this difference in structure of both oxides, the Fe and Mn translocation from root to shoot in white lupin plants is facilitated and Fe and Mn accumulation in roots is avoided when FMBO is applied. Therefore, FMBO seems to be a good choice to be applied as Fe and Mn micronutrient fertilizers; both vegetative experiments and batch experiments should be conducted to confirm these preliminary findings as already reported elsewhere. Batch experiments were conducted to check how synthetic iron chelates (Fe-EDDHA, Fe-EDDHMA and Fe-HBED) could remain significantly adsorbed onto two-line ferrihydrite thereby notably affecting their effectiveness as iron micronutrient fertilizer [61–64]. Effectiveness of synthetic siderite ($FeCO_3$) in preventing Fe chlorosis in olive was demonstrated by conducting three-year experiments in three orchards [63].

## 5. Conclusions

Synthetics AMO and FMBO were evaluated as heavy metal ($Ag^+$ and $Tl^+$) absorbents and as ecofriendly fertilizers in micronutrients (Fe and/or Mn) for white lupin plants. AMO, an amorphous mixture of Mn oxides, and FMBO, an Fe dopped birnessite, were shown to be efficient adsorbents due to their high superficial area and stability. In the presence of white lupin plants, in a slightly acidic nutrient solution, AMO interacted with the FeEDDHA iron chelate producing Fe and Mn accumulation in roots, due to the FeEDDHA adsorption onto AMO surface. Moreover, EDDHA may release Fe and chelated Mn by increasing the white lupin Mn uptake. Conversely, FMBO released high contents of Mn after interactions with $Ag^+$ or $Tl^+$ solutions at a similar pH to that of the nutrient solution where white lupin plants were grown, but they did not show Fe or Mn accumulation in roots or toxicity symptoms in shoots in the presence of FMBO. As already discussed, further research should be conducted to confirm the potential use of these metal (hydr)oxides as sorbents materials and micronutrient fertilizers. AMO and FMBO not only have demonstrated to be efficient adsorbents of emergent heavy metals, but also could be promising materials to be taken into consideration in Fe and Mn nutrition.

**Author Contributions:** Conceptualization: F.Y, M.T.C. and E.E. Writing—original draft preparation: M.T.C.; NMOs synthesis and characterization: M.d.F., J.C., M.T.C. and F.Y.; heavy metals interaction and bioassay: P.H. and E.E.; Mössbauer spectroscopy: J.S.-M.; supervision: E.E. and F.Y.; project

administration: F.Y.; funding acquisition: J.J.L. All authors have read and agreed to the published version of the manuscript.

**Funding:** This research was funded by the Spanish State Research Agency, Spanish Ministry of Science, Innovation and Universities (project RTI2018-096268-B-I00) and the Comunidad de Madrid (Spain) through the Structural Funds 2014-2020 (ERDF and ESF) for financial support (project AGRISOST-CM S2018/BAA-4330).

**Institutional Review Board Statement:** Not applicable.

**Informed Consent Statement:** Not applicable.

**Data Availability Statement:** Not applicable.

**Conflicts of Interest:** The authors declare no conflict of interest. The funders had no role in the design of the study; in the collection, analyses, or interpretation of data; in the writing of the manuscript, or in the decision to publish the results.

## Abbreviations

| | |
|---|---|
| AMO | amorphous Mn oxide |
| FMBO | Fe-Mn binary oxide |
| 2L-Fh | two-line ferrihydrite |
| Fe-EDDHA | Fe (III) ethylenediamine bis (o-hydroxyphenyl acetic acid) |
| Fe-EDDHMA | (Fe (III) ethylenediamine di-(o-hydroxy-p-methyl-phenylacetic) acid), iron (III) N,N′-bis(o-hydroxybenzyl)-ethylenediamine-N,N′-diacetic acid |
| FTIR | Fourier transform infrared spectroscopy, ICP MS: inductively coupled plasma mass spectrometry, XRD X-ray diffraction |
| ANOVA | analysis of variance |

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
