# Peer review of "Synthesis and Characterization of Nano Fe and Mn (hydr)oxides to Be Used as Natural Sorbents and Micronutrient Fertilizers"

_agronomy, doi:10.3390/agronomy11091876_

Round 1

Reviewer 1 Report

The authors' research is important from a practical point of view and can form the basis for further research. However, it seems to me that when writing for an agricultural journal, the reaction of plants to the studied chemical compounds should be described in more detail. For example, the developmental stage at which the plants were harvested was not indicated. The value of the article would be also increased by statistical and graphical presentation of accumulation and content of the studied elements as well as dry mass in particular parts of plants. The above remark does not disqualify the article. However, I think that the article should not be published without improving subsection 27 Statistics and improving Figures 5-7. Detailed comments in annex

Reviewer 2 Report

Title: Synthesis and characterization of nano Fe and Mn hydr(oxides) for environmental and agricultural uses
Abstract: This section requires complete revision to make it concise and a thorough English revision. The section should encompass the entire study (introduction, aim, hypothesis, result and conclusion). Could you also clearly explain abbreviations (e.g. Fe-EDDHA) in the section?
Introduction:
I find this section fairly good. I will only suggest the aim of the study should be made evident in this section especially in the beginning and concluding parts. 
Materials and Methods: 
 I find this section well-structured and scientifically valid. I will only suggest some description of the study area is provided. 
Results: The results are well presented. 
Discussion: This section needs thorough revision and require more justifications with some more comparisons with similar studies in the past.
The outcome of the study “Synthetics AMO and FMBO were evaluated as heavy metal (Ag+ and Tl+) absorbents and as ecofriendly fertilizers in micronutrients (Fe and/or Mn) for white lupin plants” is very important especially in this period of uncontrolled soil and water pollution due mining, conventional farming etc.  
Although, the study is interesting and will appeal to readers, I suggest a thorough revision of the discussion section is done to bring the quality to a publishable level. Also a through grammar revision should be done.

Reviewer 3 Report

In this paper, nanomaterials of copper and manganese were synthesized and their effects on plant growth and development were studied, which has important research value in agriculture. However, further changes are needed before publication. 1, The title of the article is too big. What is the application of nanomaterials in agriculture? Fertilizer? pesticides? Or control soil pollution? 2, The latest references on many applications of nanomaterials in agriculture (fertilizers, pesticides, treatment of soil heavy metals) are not cited and discussed enough. Carbon-based nanomaterials suppress tobacco mosaic virus (TMV) infection and induce resistance in Nicotiana benthamiana. JOURNAL OF HAZARDOUS MATERIALS, 2021, 404: 124167. Physiological impacts of zero valent iron, Fe3O4 and Fe2O3 nanoparticles in rice plants and their potential as Fe fertilizers. ENVIRONMENTAL POLLUTION, 2021, 269: 116134. Application of Nanoparticles Alleviates Heavy Metals Stress and Promotes Plant Growth: An Overview. NANOMATERIALS, 2021, 11(1): 26. Alleviation of nitrogen stress in rice (Oryza sativa) by ceria nanoparticles. ENVIRONMENTAL SCIENCE-NANO, 2020, 7(10): ‏ 2930-2940. Physiological and biochemical response of wheat (Triticum aestivum) to TiO2 nanoparticles in phosphorous amended soil: A full life cycle study. JOURNAL OF ENVIRONMENTAL MANAGEMENT, 2020, 263: 110365. 3, The descriptions of materials and methods are not detailed enough, such as ICP-MS manufacturer, instrument model and instrument parameters. 4, Why are there no photos of shoots or whole plants in Figure 8? Especially the data of plant height, dry weight, root length and root weight and their significance analysis are very important.

Round 2

Reviewer 2 Report

The author has modified the manuscript as per my suggestion. I suggest accepting the manuscript in its present form.